# Observing collisions beyond the secular approximation limit

Junyang Ma[1,2], Haisu Zhang[1], Bruno Lavorel[1], Franck Billard[1], Edouard Hertz [1], Jian Wu[2,3]*, Christian Boulet[4], Jean-Michel Hartmann[5]* & Olivier Faucher [1]*

Quantum coherence plays an essential role in diverse natural phenomena and technological applications. The unavoidable coupling of the quantum system to an uncontrolled environment incurs dissipation that is often described using the secular approximation. Here we probe the limit of this approximation in the rotational relaxation of molecules due to thermal collisions by using the laser-kicked molecular rotor as a model system. Specifically, rotational coherences in $N_2O$ gas (diluted in He) are created by two successive nonresonant short and intense laser pulses and probed by studying the change of amplitude of the rotational alignment echo with the gas density. By interrogating the system at the early stage of its collisional relaxation, we observe a significant variation of the dissipative influence of collisions with the time of appearance of the echo, featuring a decoherence process that is well reproduced by the nonsecular quantum master equation for modeling molecular collisions.

---

[1] Laboratoire Interdisciplinaire CARNOT de Bourgogne, UMR 6303 CNRS-Université de Bourgogne Franche-Comté, BP 47870, 21078 Dijon, France. [2] State Key Laboratory of Precision Spectroscopy, East China Normal University, 200062 Shanghai, China. [3] Collaborative Innovation Center of Extreme Optics, Shanxi University, 030006 Taiyuan, Shanxi, China. [4] Institut des Sciences Moléculaires d'Orsay, CNRS, Université Paris-Sud, Université Paris-Saclay, 91405 Orsay, France. [5] Laboratoire de Météorologie Dynamique/IPSL, CNRS, Ecole polytechnique, Institut polytechnique de Paris, Sorbonne Université, Ecole Normale Supérieure, PSL Research University, 91120 Palaiseau, France. *email: jwu@phy.ecnu.edu.cn; jmhartmann@lmd.polytechnique.fr; olivier.faucher@u-bourgogne.fr

**E**nergy transfers through quantum coherences are key mechanisms in natural phenomena and technological applications, such as human vision[1], light-harvesting complexes[2], quantum heat engines[3], and quantum information and computing[4]. The understanding of the long-lived coherence involved in these phenomena requires a detailed modeling of the system-bath interactions beyond the so-called secular and/or Markovian approximations[5,6]. Despite continuous theoretical progress on understanding nonsecular dynamics in the last decades, convincing experimental observations are still lacking, in particular when they manifest through collision-induced processes. More generally, understanding how collisions between molecules change their translational and rotational motions is a fundamental problem of physics that has numerous practical implications. Indeed, most real gas media being at non-negligible pressure, intermolecular interactions affect many observable quantities. At thermodynamic equilibrium, they play a key role in the shape of light-absorption and light-scattering spectra[7] and in the transports of mass and energy[8,9], for instance. When the system has been brought out of equilibrium by some external action (for example one or several resonant or nonresonant electromagnetic excitations or a plasma discharge), collisional processes are often the main channel inducing decoherence and transfers among populations bringing it back to equilibrium[8,10]. Collisional effects have been extensively studied in the spectral domain by looking at the shape of various absorption features[7]. In the time domain, it has been recently shown that the decay of alignment echoes induced by two laser kicks[11] permits time-resolved measurements of collision-induced rotational changes at short time scales[12,13]. So far, all quantum models used for the calculation of the effects of decoherence processes due to intermolecular forces are based upon the Markovian and secular approximations[8,14–16]. These allow to obtain the density matrix $\boldsymbol{\rho}(t)$ describing the evolution of the system when coherences (i.e., nondiagonal elements of $\boldsymbol{\rho}$) have been created (for example by laser pulse(s), as in the present work). Recall that the Markov approximation assumes[14,15] that all collisions are complete and un-correlated (no memory effects in the system-bath history) while the secular approximation assumes[16] that the delay $\Delta t$ at which the system is observed after the creation of the coherences is much greater than the periods of oscillation of the coherences.

Here we provide the experimental and theoretical evidence that rotational alignment echoes enable to probe an open molecular system at the early stage of its collisional relaxation where the secular approximation breaks down. These echoes[11] occur after molecules have been suddenly aligned by two successive nonresonant laser excitations. A first short pulse, $P_1$, impulsively aligns some of the molecules along its polarization direction thanks to the anisotropy of the molecular polarizability[17]. This results in a peak in the alignment factor $\langle\cos^2(\theta)\rangle$ $(t)$ ($\theta$ being the angle between the molecule axis and the laser polarization direction and $\langle\rangle$ denoting the quantum expectation value) that quickly vanishes due to rotational dephasing. After a delay $\tau_{12}$, a second pulse, $P_2$, is applied that induces a rephasing process leading to the creation of an echo in the alignment factor at time $\Delta t = 2\tau_{12}$ after $P_1$. The decay, with pressure, of the echo is studied in $N_2O$–He gas mixtures for various delays $\tau_{12}$. The experiments show that the dissipation of the echo is all the more rapid when the delay is long, a result in contrast with the secular model predicting delay-independent results. On the other hand, very good agreement between measurements and calculations is obtained when a nonsecular approach is used.

## Results

**Experiments**. The experiment is implemented with the setup depicted in Fig. 1 and further described in the Methods section. After the molecules have been aligned by two laser kicks, the anisotropy of the medium resulting from the reorientation of the molecules is interrogated by a probe pulse. A detection is performed so that the measured signal is directly proportional to the convolution of $(\langle\cos^2(\theta)\rangle(t) - 1/3)$ with the temporal envelope of the probe, where 1/3 is the alignment factor obtained when molecules are randomly oriented. Figure 2a exemplifies the echo features observed in the alignment signal of linear molecules at low pressure. Like any other echo phenomenon, the dominant echo response appears at the symmetric position of $P_1$ with respect to $P_2$ along the time axis. It is followed by higher-order and imaginary echoes[18,19] and revivals. All these structures exhibit an asymmetric temporal shape consisting of a peak resulting from the alignment of the molecules along the polarization of the pump field, and a dip associated with a delocalization of the molecular axes within a plane perpendicular to the field vector.

It is only very recently that the collisional dissipation of echoes has been studied[12,13], by analyzing the evolution of their amplitudes for various delays at fixed gas density. The interest of this approach to probe the dissipation of the alignment under high-pressure conditions where standard alignment revivals cannot be used was demonstrated in ref. [12]. Indeed, revivals restrict the probing of the system to specific times that are tied to the moment of inertia of the molecule. On the contrary, the rotational echo provides much more flexibility, since its time of appearance can be tuned at will by adjusting the delay between $P_1$ and $P_2$. The echo is therefore particularly well suited for interrogating the dynamics of the system at early times, much before the first revival appears. As shown in Fig. 2b, we here look at the echoes from a point of view different from that used in refs. [12,13], and study the reduction of their density-normalized amplitudes "$S$" with increasing pressure for fixed values of $\tau_{12}$. Note that all results presented from now on have been recorded in high-pressure (up to 25 bar) mixtures of $N_2O$ gas diluted in 96% of helium. The high concentration of helium, which has been chosen for its weak nonlinear optical properties, allows overcoming the nonlinear propagation effects that would take place in high-density pure molecular gases of the same nature. A first indication of the enriched information brought by the echo with respect to the revivals is revealed by Fig. 2c, which shows measured amplitudes of five echoes and two revivals as a function of the density $d$ multiplied by the time of appearance of the considered alignment structure (i.e., the echo or the $n$th revival). Obviously, the efficiency with which collisions reduce the alignment amplitude is the same for the half and full revival (at 20.2 and 40.4 ps), while it varies significantly for the echoes observed between $\tau_{12} = 1.61$ and 8.58 ps. By repeating the measurements of Fig. 2c for several delays and gas densities, we have been able to extract (see the Methods section) density-normalized characteristic time constants of the echo amplitude decay from exponential fits (see Fig. 2c). The values reported in Fig. 3 show that the echo decay is slow at short times and becomes faster as the delay increases, before a plateau is reached around $2\tau_{12} = 10$ ps. In the plateau region, the echo and the revival share about the same density-normalized characteristic decay time. Note, for comparison, that the mean time interval between successive $N_2O$–He collisions, obtained for the intermolecular distance of 3.5 Angstrom where the $N_2O$–He potential is significant (see Supplementary Note 1), is of 70 ps.amagat.

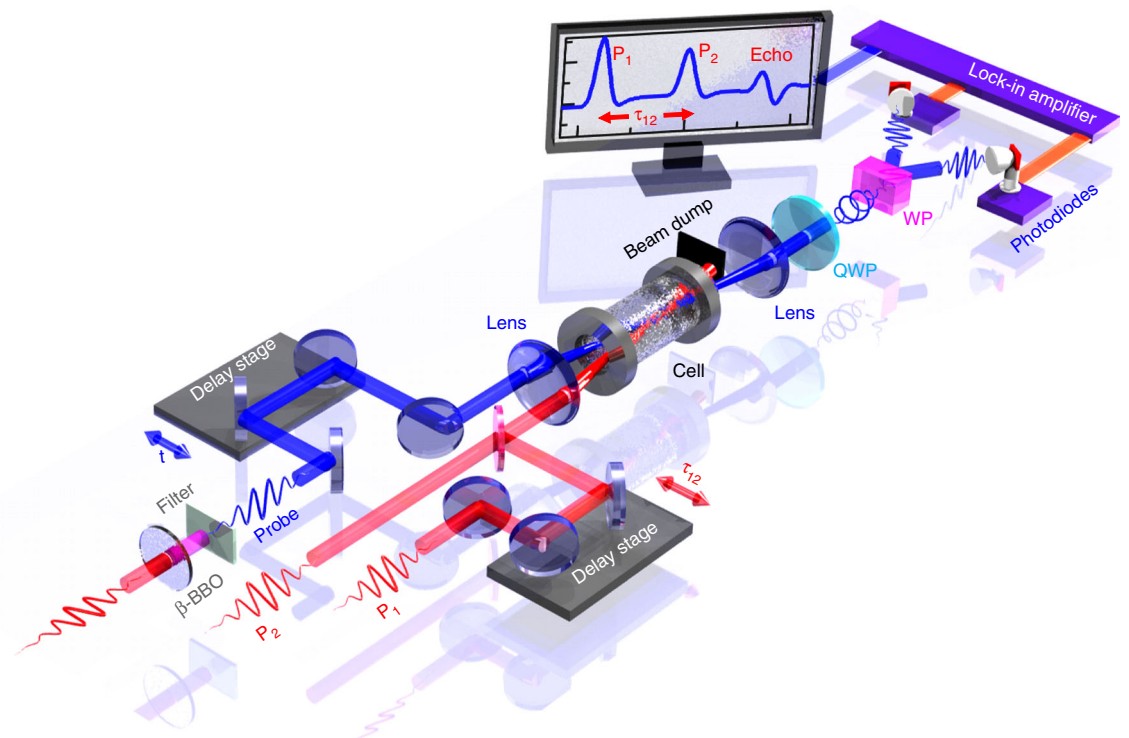

**Fig. 1 Schematic representation of the experimental set-up.** $N_2O$ gas molecules contained in a room-temperature high-pressure gas cell are impulsively aligned by two time-delayed 800 nm pulses $P_1$ and $P_2$. The induced rotational dynamics is measured through the time-dependent birefringence experienced by a 400 nm probe pulse. The detection uses two photodiodes connected head-to-tail to a lock-in amplifier delivering a signal proportional to the part of the probe field that has been depolarized by the aligned molecules. The image reproduced on the computer screen illustrates the alignment echo signal produced by the two time-delayed strong laser kicks; QWP, quarter-wave plate; WP, Wollaston prism.

**Model**. In order to explain the difference between the short-time and long-time dynamics, we performed numerical simulations of the alignment signal in the presence of collisional dissipation. Starting from equilibrium before the pulses, where the density operator $\rho(t < 0)$ is Boltzmannian and diagonal with respect to the rotational quantum numbers $J$ (principal) and $M$ (magnetic), the evolution for $t \geq 0$ is obtained from the Liouville-von Neumann equation[8,14–16]

$$\frac{d\rho}{dt}(t) = -\frac{i}{\hbar}[\mathbf{H}_0 + \mathbf{H}_L(t), \rho(t)] + \left(\frac{d\rho(t)}{dt}\right)_{Coll}, \quad (1)$$

where $\hbar$ is the Planck constant divided by $2\pi$, $\mathbf{H}_0$ is the free rotational Hamiltonian, and

$$\mathbf{H}_L(t) = -\frac{1}{4}E^2(t)\Delta\alpha\cos^2\theta \quad (2)$$

describes the nonresonant interaction of the molecule with the linearly polarized field (up to an irrelevant term that does not multiply the direction cosine operator $\cos^2\theta$), with $\Delta\alpha = \alpha_{//} - \alpha_{\perp}$ the anisotropic polarizability and $E(t)$ the temporal envelop of the laser pulse. Note that, in principle, the contribution of the permanent dipole moment of $N_2O$ should be included in Eq. (2). However, since the excitation field used in the experiment (at 800 nm) oscillates with a period (2.7 fs) much smaller than that of the rotation of the molecule for the thermally populated levels, the associated contribution to Eq. (2) averages out. Assuming a Markovian model of collisions (valid here because, as discussed in the Supplementary Note 1, $N_2O$ and He interact very shortly and in a very limited range of intermolecular distance[20]), and neglecting all radiative decays since they occur on a much longer time scale, the matrix elements of the dissipation

term read

$$\left(\frac{d\rho_{ij}(t)}{dt}\right)_{Coll} = -d\sum_{i'j'}\wedge_{ij,i'j'}\rho_{i'j'}(t), \quad (3)$$

where $i$ and $j$ denote the rotational states $|J,M\rangle$ of the system and $\Lambda_{ij,i'j'}$ are the density-normalized relaxation matrix elements.

**Break down of the secular approximation**. In a first step, we use the so-called Bloch model in which dissipative effects are treated within the Markovian and secular approximations (further discussed in the Supplementary Notes 1 and 2, respectively). This corresponds to neglecting all elements of the relaxation matrix responsible for transfers among coherences (nondiagonal terms of $\rho$) and between populations (diagonal elements of $\rho$) and coherences. Only the $\Lambda_{ij,ij}$ and $\Lambda_{ii,i'i'}$ terms are then kept which are directly related to the rates of population transfer from $|J,M\rangle$ to $|J',M'\rangle$ (see ref. [21] and Supplementary Note 3), including inelastic ($J$-changing) and elastic reorienting ($J$-conserving, $M$-changing) processes, as well as the pure dephasing of the coherence which describes the effect of elastic collisions that interrupt the phase of the molecule without quenching it. Under these approximations, the equation driving the decay of the coherences (i.e., for $i \neq j$) becomes

$$\left(\frac{d\rho_{ij}(t)}{dt}\right)_{Coll}^{Secular} = -d\Lambda_{ij,ij}\rho_{ij}(t) \quad (4)$$

The diagonal elements $\Lambda_{ij,ij}$ have been estimated for $N_2O$–He at room temperature (see the Supplementary Note 3.1) using the infinite order sudden approximation (IOSA). The calculations show that all these terms are positive, practically independent of $i$ and $j$ and equal to 0.013 $ps^{-1}$ amagat$^{-1}$. This

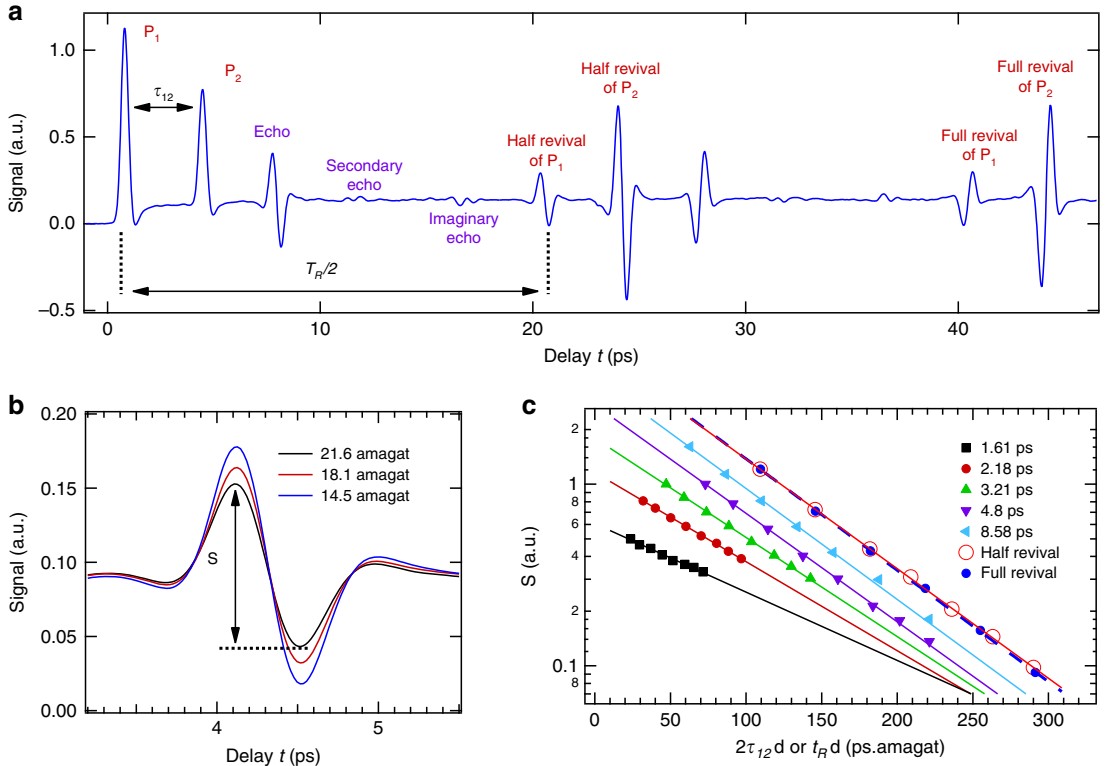

**Fig. 2 Rotational structures of N₂O aligned by two laser kicks. a** Alignment signal recorded in pure $N_2O$ gas at low pressure by scanning the temporal delay between the first aligning pulse $P_1$ (at $t = 0$) and the probe pulse over slightly more than the full rotational period $T_R = 40.4$ ps of the molecule. The peaks identified by $P_1$ and $P_2$ correspond to the transient alignment signals produced by the two pulses separated by the delay $\tau_{12}$. The main echo is generated at $t = 2\tau_{12}$, with the secondary echo observable at $t = 3\tau_{12}$, and the imaginary echo produced at $T_R/2 - \tau_{12}$ (equivalent features also appearing at times shifted by $+T_R/2$). In addition to echoes, other transients corresponding to the standard half and full alignment revivals of $P_1$ (at $T_R/2$ and $T_R$) and $P_2$ (at $T_R/2 + \tau_{12}$ and $T_R + \tau_{12}$) are also observed. **b** Alignment traces of $N_2O$ diluted in He measured at various densities around the main echo at $2\tau_{12}$ for $\tau_{12} = 2.18$ ps. The amplitudes $S$ of the alignment structures are measured from peak to dip. **c** Amplitudes of the half revival (open red circles), full revival (full blue circles), and of the main echo for five different values of the delay $\tau_{12}$ versus the gas density $d$ multiplied by the time of observation $t_R$ ($T_R/2$ or $T_R$) and $2\tau_{12}$ for the revivals and echo, respectively, expressed in picosecond amagat (ps.amagat, with 1 amagat = $2.687 \times 10^{25}$ mol. m$^{-3}$) units. The lines indicate the best exponential fits.

means that the echo decay within the secular approximation is independent of the delay. As shown by the simulations represented by the black dashed line in Fig. 3, the secular theory well reproduces the observed decays of the echoes in the plateau region beyond about $2\tau_{12} = 10$ ps with a density normalized time constant of $1/0.013 = 76$ ps amagat. In contrast, the secular model does not capture the weaker efficiency of collisions at shorter times and the fact that the decrease of the echo amplitude with increasing density is here all the more slow when the delay is short. This observation seriously questions the validity of the secular approximation during the early dynamics of the system.

For a simple (but realistic) picture of the limits of the secular approximation, let us assume that coherences have been instantaneously generated by a laser pulse at $t = 0$. This creates nonzero nondiagonal elements $\rho_{JM,J'M}(t = 0^+)$ between states $|J,M\rangle$ and $|J',M\rangle$, which, in the absence of collisions, then evolve with time according to $\rho_{JM,J'M}(t) = \rho_{JM,J'M}(t = 0^+)$ $\exp[-i(E_J - E_{J'})t/\hbar]$, with $E_J$ and $E_{J'}$ the corresponding rotational energies. Let us now assume that collisions induce transfers between coherences and, in particular, from $\rho_{J_1M_1,J'_1M_1}(t)$ to $\rho_{J_0M_0,J'_0M_0}(t)$ with a real valued rate constant. Under some approximations presented in the Supplementary Note 2, it can be shown that the relative change of $\rho_{J_0M_0,J'_0M_0}(t)$

due to transfers from $\rho_{J_1M_1,J'_1M_1}(t)$ at time $t$ after the laser pulse is

$$\text{Re}\left[\frac{\Delta\rho_{J_0M_0,J'_0M_0}(t)}{\rho_{J_0M_0,J'_0M_0}(t)}\right] = B\text{sinc}\left[(\omega_{J_1J'_1} - \omega_{J_0J'_0})t\right], \quad (5)$$

where Re[...] denotes the real part, $B$ is a constant, sinc[...] designates a cardinal sine, and $\omega_{JJ'} \equiv (E_J - E_{J'})/\hbar$. It is obvious from Eq. (5) that, as $|\omega_{J_1J'_1} - \omega_{J_0J'_0}|t$ gets larger than a few times $\pi$, the transfers between coherences become negligible when compared to the total loss. Note that the imaginary part of $\Delta\rho/\rho$ exhibits the same behavior. In order to be more quantitative, let us consider the case of $N_2O$. Since only $\rho_{JM,J'M}(t)$ with $J' = J \pm 2$ contribute to the echo amplitude (see Methods section), we below focus on such elements and limit ourselves to the most populated state $J_0 = 15$ at $T = 300$ K. The quantity in Eq. (5) is plotted in Fig. 4a for $J_0 = 15$, $J'_0 = 17$, $J'_1 = J_1 + 2$ and $J_1 = 17, 19, 21$, and 23 (we keep $|J_0 - J_1|$ even because of the collisional selection rule, see the Supplementary Note 3.2), where all values have been normalized to unity at $t = 0$. As can be seen, transfers between coherences indeed decrease rapidly. They are significant before about 5 ps and practically negligible above 10 ps, a delay beyond which the secular approximation thus becomes valid, a behavior qualitatively consistent with the difference observed at short times between the experimental results and the secular predictions in Fig. 3.

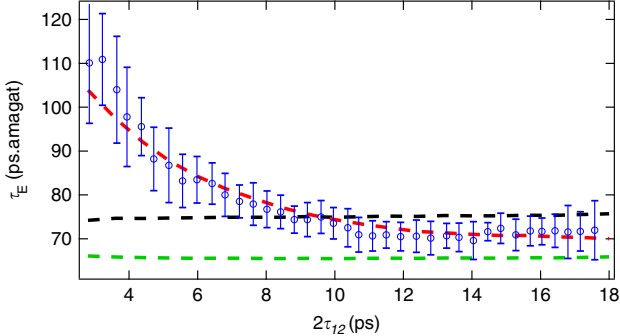

**Fig. 3 Time constants of collisional dissipation of N₂O.** The blue circles with error bars (representing two standard deviations of the mean) are the density-normalized decay time constants $\tau_E$ of the echoes deduced from the measurements of the alignment signal recorded at various $N_2O(4\%)$ + $He(96\%)$ gas densities and fixed delays $\tau_{12}$ between the two pulses. The dashed lines denote the results of simulations conducted by solving the density matrix equations for molecules impulsively aligned by two short laser pulses and interacting with each other through collisions. The green and black dashed lines have been obtained using the standard Bloch equations (i.e., using the secular approximation) with initial $N_2O$ rotational populations corresponding to temperatures of $T = 100\,K$ and $295\,K$, respectively. The red dashed line represents the results obtained, for populations associated with $T = 100\,K$ (see text), using the nonsecular Redfield equations, i.e. including all relaxation terms in Eq. (3).

**Successful nonsecular predictions.** In order to go further, computations of the echo decays have been carried by solving Eqs. (1)–(3) now keeping all collisional transfers channels and thus accounting for secular and nonsecular terms. Equation (3) thus becomes

$$\left(\frac{d\rho_{ij}(t)}{dt}\right)^{Nonsecular}_{Coll} = \left(\frac{d\rho_{ij}(t)}{dt}\right)^{Secular}_{Coll} - d\sum_{i',j'\neq i,j}\wedge_{ij,i'j'}\rho_{i'j'}(t), \quad (6)$$

where the first term on the right-hand side is given by Eq. (4). The required relaxation matrix elements for Eq. (6) have been constructed using the IOSA, as explained in the Supplementary Note 3 where the consistency with the rates used previously within the secular approximation[21–23] is demonstrated. Unfortunately, due to computer-time and computer-memory limitations, nonsecular calculations at room temperature, which require to include $N_2O$ $J$ values up to $J_{Max} = 60$, were not tractable (which is not the case when the secular approximation is used). Hence, we carried calculations for $T = 100\,K$ and $J_{Max} = 38$, conditions for which predictions for 6 densities and 21 delays $\tau_{12}$ could be completed in a "reasonable" time. The results of this exercise, in which we used rates computed at 295 K but initial populations defined at 100 K, are displayed in Fig. 3 with a red dashed line. Note that comparing the secular results at 100 K and 300 K shows that limiting computations to $T = 100\,K$ and $J_{Max} = 38$ only slightly affects the prediction (by 10%). Figure 3 confirms that the nonsecular terms [within the sum on the right-hand side of Eq. (6)], which describe the transfers among coherences as well as

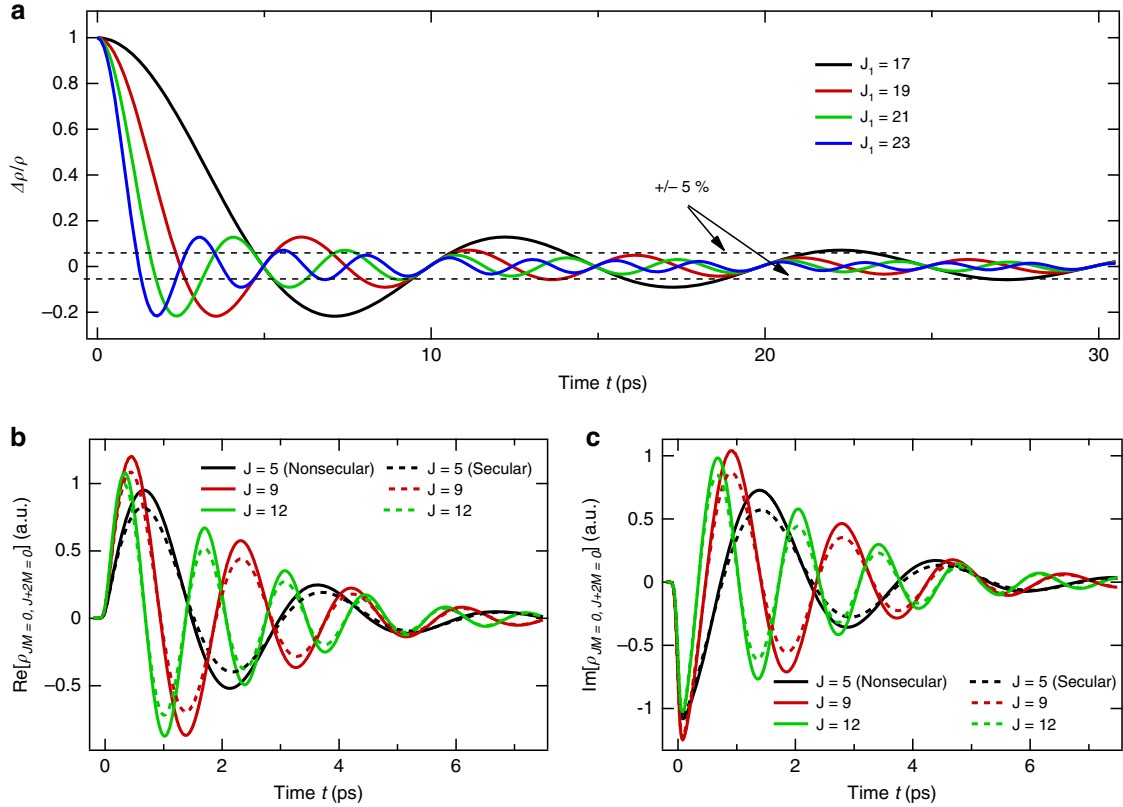

**Fig. 4 Secular and nonsecular effects. a** Relative modification of the $\rho_{J_0M_0,J_0+2M_0}(t)$ coherence for $J_0 = 15$ induced by collisional transfers from the coherences $\rho_{J_1M_1,J_1+2M_1}(t)$ with $J_1 = 17$ (black), $J_1 = 19$ (red), $J_1 = 21$ (green), and $J_1 = 23$ (blue), all normalized to unity at $t = 0$. This simple modeling (see Supplementary Note 2) of coherence transfers considering only few rotational states around the most populated state of $N_2O$ at 300 K reveals that during the short time evolution of the system, exchanges between coherences are efficient, which slow down the decay of the alignment factor with respect to what would be obtained using the secular approximation that neglects these collisional transfers and only takes the losses into account. **b, c** Computed real and imaginary parts of the $\rho_{JM=0,J+2M=0}(t)$ coherences for $N_2O$ in He at 33 amagat, obtained using the nonsecular (full lines) and secular (dashed lines) models.

between populations and coherences, globally contribute to reduce the pressure-induced echo decay at short times. Analysis shows that this results from the fact that these contributions here systematically reduce the collision-induced decay of the amplitudes of the $\rho_{JM,J\pm2M}(t)$ coherences, as exemplified in Fig. 4b, c, and thus also of the echo. In particular, for $N_2O$ diluted in He, the IOS model predicts that, immediately after the first pulse, nonsecular couplings between coherences reduce the collisional damping of these amplitudes by a factor of about two with respect to the secular behavior. Neglecting the nonsecular terms thus leads to overestimate the decay of the system. Note that a similar pattern was observed in the ref. [24] in the modeling of decoherence and dissipation in non adiabatic *cis–trans* photoisomerization. As time goes by, the averaging out of these terms, due to progressive dephasing of the coherences (see Fig. 4a), results in a decrease of the predicted decay time constant toward the secular plateau, in perfect agreement with the measurements. Note that the convergence of the nonsecular results toward the delay-independent secular ones (Fig. 3) explains the experimentally observed close similarity of the decays of the revivals and of the echo for large delays (Fig. 2c).

## Discussion

Collisional dynamics of rotationally excited $N_2O$ molecules diluted in helium gas has been investigated using alignment echoes. The latter enable to probe the system very soon after its stimulation by two laser kicks, on a time scale of a few ps that is comparable to the time interval between successive collisions (about 3.5 ps for the density of 20 amagat typical of the experiments) at the high pressures used in the measurements. This unprecedented scrutiny enables the observation of collisional transfers occurring in the nonsecular regime and leading to a slowing-down of the decoherence of the system lasting for a few picoseconds. This observed longevity of coherences challenges the traditional wisdom that interactions with the environment universally lead to decoherence only, and boosts the theoretical revisiting of the quantum master equations normally used to describe the system-bath interactions. It also opens renewed perspectives, through pure time-domain experiments, for our understanding of collisional processes and tests of rotational relaxation models. In particular, probing echoes at short times for collisional pairs involving significantly smaller relative speeds and longer-range intermolecular forces that does the $N_2O$–He pair studied in this work should enable to evidence the breakdown of the Markov approximation. Finally, note that equivalent studies could be made for other types of molecules (including polar and nonlinear ones) and/or carried out by using a different process in order to excite the system at $t=0$ and $t=\tau_{12}$ and generate an echo at $2\tau_{12}$. For instance, early time decoherence could be investigated for $CH_3I$ using THz excitations as done in the refs. [25–27] with a single pulse, which is only one example among many others of the great potentialities of the approach introduced in this paper to study early time dissipation.

## Methods

**Experimental set-up.** The experiment is based on a chirped femtosecond Ti:Sapphire amplifier delivering 800 nm light pulses with a repetition rate of 1 kHz and a duration of 100 fs (FWHM). The output of this laser is first separated into a pump and a probe beam. The pump beam, used for impulsive alignment, is directed through a Mach-Zehnder interferometer producing two linearly-polarized collinearly-propagating pulses $P_1$ and $P_2$ with a time delay controlled by a motorized stage. In order to reduce the perturbation of the detection by the light of the pump pulses scattered by the molecules and the cell windows, the probe beam is frequency doubled with a type I BBO crystal before being sent through a second motorized delay line. For the purpose of the birefringence detection, its polarization is rotated by 45° with respect to the pump pulses. The pump and probe beams are both focused in a high-pressure static cell by the same lens ($f=100$ mm) in a noncollinear geometry leading to a crossing angle of 4° at focus. The intensities of $P_1$ and $P_2$ are estimated around 20 and

13 TW cm$^{-2}$, respectively. After filtering out the 800 nm photons and collecting the 400 nm light with a second lens, the transmitted probe light is analyzed by a low noise balanced detection. The latter is achieved by combining a quarter-wave plate and a Wollaston prism separating the vertical and horizontal polarization components that are then independently measured by two connected head-to-tail photodiodes. All measurements were made at room temperature (295 K) for 4%$N_2O$+96%He mixtures at various total pressures up to 25 bar.

**Computations.** The density matrix $\rho$ was initialized, before the first laser pulse, to its equilibrium value at 100 and 295 K (see text). Its evolution with time was then obtained by solving Eqs. (1) and (2) for a 100 fs Gaussian pulse envelop (FWHM) and the same pulses intensities as in the experiments, using the anisotropic polarizability[28] $\Delta\alpha=19.8$ $a_0^3$ and the rotational constant $B=0.419$ cm$^{-1}$ of $N_2O$. The collisional term was computed using Eq. (3), with rates $\Lambda_{J_1M_1,J_1'M_1'\rightarrow J_0M_0,J_0'M_0'}$ constructed as explained in the Supplementary Note 3. Considering that all operators are diagonal in $M$, only the $\langle JM|\rho(t)|J'M\rangle$ terms were computed. From knowledge of these density matrix elements, the alignment factor was computed from

$$\langle\cos^2\theta-1/3\rangle=\sum_{J,M}\langle JM|\rho(t)\cos^2\theta|JM\rangle=\sum_{J,M,J'}\langle JM|\rho(t)|J'M\rangle\langle J'M|\cos^2\theta|JM\rangle$$

(7)

with $\langle J'M|\cos^2\theta|JM\rangle=\frac{2}{3}\sqrt{(2J+1)(2J'+1)}\begin{pmatrix}J&2&J'\\0&0&0\end{pmatrix}\begin{pmatrix}J&2&J'\\M&0&-M\end{pmatrix}$, where (:::) is a 3 J symbol. Note that this implies $J'-J=0\pm2$.

**Data analysis.** For each delay $\tau_{12}$ between the laser pulses $P_1$ and $P_2$, the measured and computed time-dependent alignment factors for various gas densities $d$ were analyzed as follows. The variation with $d$ of the peak-to-dip density-normalized amplitude $S$ of the echo (see Fig. 2b) was (nicely, see Fig. 2c) least-squares fitted by $S(d,\tau_{12})=A(\tau_{12})\exp[-d/d_0(\tau_{12})]$ floating $A(\tau_{12})$ and $d_0(\tau_{12})$. A density-normalized decay time constant was then defined as $\tau_E(\tau_{12})=2\tau_{12}d_0(\tau_{12})$, considering the fact that the echo appears at $t=2\tau_{12}$ after the time origin defined by $P_1$. For the revivals, the same approach was used except that $P_2$ was turned off, and a density-normalized time constant was determined for each of them from the decay of their amplitude with density using $\tau_R(t_R)=t_Rd_0(t_R)$, where $t_R$ is the time delay between the revival and $P_1$.

## Data availability

The measured and simulated data that support the findings of this study are available on reasonable request from O. Faucher and J.-M. Hartmann, respectively.

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

## Acknowledgements

The work was supported by the ERDF Operational Program—Burgundy 2014/2020 and the EIPHI Graduate School (contract "ANR-17-EURE-0002"). J.M. acknowledges the support from the China Scholarship Council (CSC). J.-M.H. benefited, for the computer simulations, from the IPSL mesocenter ESPRI facility which is supported by CNRS, UPMC, Labex L-IPSL, CNES, and Ecole Polytechnique. J.W. acknowledges the support by the National Key R&D Program (Grant No. 2018YFA0306303) and the NSFC (Grant Nos. 11425416, 11761141004, and 11834004).

## Author contributions

O.F. with J.-M.H. planned the project and wrote the manuscript. H.Z. and B.L., with the assistance from J.M. and F.B., designed and installed the experiment. J.M. carried out the measurements. B.L. and J.M. analyzed the data. J.-M.H. and C.B. developed the theory and carried out the simulations. E.H. and J. W. discussed the results and the manuscript with the rest of the authors.

## Competing interests

The authors declare no competing interests.
