## [Peer Review File · Nature Communications]

Reviewers' comments:

Reviewer #1 (Remarks to the Author):

In this manuscript the authors describe an experiment that tracks the ultrafast decoherence of an open quantum system. The experimental results are interpreted by supporting theoretical simulations implemented by numerical propagation of the Liouville-Von Neuman equation. Previously derived expressions for relaxation matrix elements are used, of which coherence transfer matrix elements are excluded within the secular approximation. The specific system interrogated is a coherently excited rotational wavepacket in a pressurized, room temperature gas of Helium seeded with N₂O. Experimentally measured amplitudes of rotational echoes induced in the N₂O serve as a probe of the collisional system-bath coupling. The use of echoes as a probe of decoherence was previously established in ref. 8, where the dissipation of coherence in CO₂ was tracked by measuring the amplitude of the echoes as a function of delay between the pulse pair inducing the echoes.

While the study in ref. 8 probes similar information with regards to rotational wavepackets, it is clearly a preliminary study in which the decay of the echoes is modeled classically. The present study however is a rigorous, quantum mechanical study of the loss of coherence. This is a fundamentally important problem that is often difficult to access experimentally. The work presented in this manuscript is therefore correspondingly interesting, and of fundamental value. From the perspective of molecular physics, one could easily envision applications of these ideas to the study of coherence transfer via intramolecular couplings, which are often treated phenomenologically under the secular approximation, i.e. assuming a single decay constant for each coupling process without considering the dynamics of each coherence. Potentially, the same procedure applied here could apply to intramolecular vibrational energy transfer by rovibrational coupling (*The Journal of Physical Chemistry* 100.31 (1996): 12735-12756), or rotational-electronic coherence transfer (*Faraday discussions* 194 (2016): 463-478, *Physical review letters* 111.24 (2013): 243005).

Given the broad interest and potential applicability of the ideas presented in this manuscript, its publication in *Nature Communications* is recommended after the following points of interest are addressed.

Coherence transfer

Within the secular approximation it is clear that each coherence decays with a time constant related to the corresponding relaxation matrix element. While the behavior of coherence transfer as a function of time has been discussed, it is difficult to envision the time dependent behavior of a single coherence. It would be illuminating to see a plot of one of the coherences addressed in Fig. 4 as a function of time. Perhaps this could be included as a second panel in Fig. 4. Further discussion of the loss of coherence due to a high rate of transfer at early times is also required. If I understand eq. (1) correctly, transfer does not imply loss, so how does enhanced transfer lead to loss?

Density-Normalized Characteristic Time Constant

The time constant as a function of delay extracted from the experimental and simulated data show that coherence transfer is important at early times. However, no direct connection is made between this time constant and the relaxation matrix, or any other fundamental theoretical parameter. Can the authors comment on the connection between this measured quantity and a parameter of the quantum system?

Reviewer #2 (Remarks to the Author):

The manuscript by J.Ma et.al provides a clear and elegant experimental observation for the non-markovian collisional relaxation of molecular rotational motion. The authors show that on short time

scales (<10ps) the secular / markovian treatment for the collision-induced decay is incorrect and that the "collision efficiency" is lower than at longer time scales in which the "memory" is lost and the secular/markovian approximation is valid. These findings show that at short time scales the effect of collisions is very different from the common treatment of collisional relaxation done in the past. Rotational dynamics on such short time scale (that are much shorter than the rotational period) were previously unattainable in rotational dynamics experiments (with which the authors have long experience and numerous important contributions to) and have become feasible only recently with the development of rotational echo measurements (a technique that some of the authors are among its pioneers [e.g. PRL. 114. 153601 (2015)]).

The topic is very timely, the results are uniquely original and utilize novel methodologies. The experimental results are clean and very persuasive. The results are clearly presented and are fully supported by state-of-the-art numerical and theoretical calculations including an elaborated SI section.

It is with great pleasure that I support acceptance and publication of this work in Nature Communications.

Few comments:

- equation (2): the authors probably meant to write $\cos^2(\theta)$ without the $-1/3$.
- It would aid the readers if the authors note that they refer to the delay of 4.8 ps when they discuss the short time scale of up to 10 ps.
- The authors write: "As shown by the simulations represented by the black line in Fig. 3, the secular theory well reproduces the observed decays of the echoes in the plateau region beyond 10 ps, but does not capture the weaker efficiency of collisions and the variation of the echo decay at short times." - I would suggest that the authors expand this very central statement by walking the readers through the results.
- last sentence of the first SI paragraph: 'amond' should be 'among'

Suggestions/questions (for the authors to decide whether to address or not)

- If possible, it would be nice to include an 'easy to understand' classical explanation about the nature of the non-Markovian collisions, i.e. how does the collision of the N₂O with the Helium atoms creates "memory effects in the system-bath "history"".
- Does the coherence amplitude (namely if the experiment is conducted with different intensities) play a role? will the memory effect hold for longer/shorter time scales? or maybe not affected at all?

Reviewer #3 (Remarks to the Author):

The paper by Faucher et al is certainly an interesting and very useful method to investigate rotational coherences on very short time scales, especially at time scales of collisions. The emphasis on studying nonsecular effects as compared to previous standard relaxation methods is informative. The paper is therefore publishable following clarification and additional information.

1- the laser intensities used are of the order of 10^{13} Watts/cm². Such intensities do produce important Stark shifts. The magnitude of these shifts is never stipulated and not clearly identified as relevant or not.

2- at the pressures used, the time for collisions is not clearly identified

3- The laser-molecule interaction assumes no dipole moment. Thus can only be applied to symmetric molecules.??

4- as shown previously by Matta et al in J Chem Phys 121,7764(2006) a permanent dipole moment and polarizability correspond to a total HL comprised of a static term and a $w, 2w$ time dependent interaction. Since the pump is at 800nm and the probe at 400nm, this would suggest interference effects from both pulses in the polarizability. Has this been observed.?

5-The imaginary part of equn 4 would contain relevant Stark and collisional energy shifts.has this been included in the simulations ?

Reasonable answers to these questions will make the paper more readable.

Replies to the reviews comments on the paper
Observing collisions beyond the secular approximation limit
submitted for publication in Nature Communications

Before detailing the changes made to the paper following the reviewers' remarks, the authors wish to thank the reviewers for their positive opinion on this work and for their constructive remarks which have enabled to significantly improve the presentation of the results. Also note that, for consistency with the Nature Communication requirements, the abstract (which contained references in our original submission to another journal) and introduction have been reorganized without any significant change in their overall content.

The comments calling for an answer appear in the revised manuscript in blue characters, our answers being in red with the **changes/adds to the text in bold characters**.

Reviewer #1 (Remarks to the Author):

In this manuscript the authors describe an experiment that tracks the ultrafast decoherence of an open quantum system. The experimental results are interpreted by supporting theoretical simulations implemented by numerical propagation of the Liouville-Von Neuman equation. Previously derived expressions for relaxation matrix elements are used, of which coherence transfer matrix elements are excluded within the secular approximation. The specific system interrogated is a coherently excited rotational wavepacket in a pressurized, room temperature gas of Helium seeded with N₂O. Experimentally measured amplitudes of rotational echoes induced in the N₂O serve as a probe of the collisional system-bath coupling. The use of echoes as a probe of decoherence was previously established in ref. 8, where the dissipation of coherence in CO₂ was tracked by measuring the amplitude of the echoes as a function of delay between the pulse pair inducing the echoes.

While the study in ref. 8 probes similar information with regards to rotational wavepackets, it is clearly a preliminary study in which the decay of the echoes is modeled classically. The present study however is a rigorous, quantum mechanical study of the loss of coherence. This is a fundamentally important problem that is often difficult to access experimentally. The work presented in this manuscript is therefore correspondingly interesting, and of fundamental value. From the perspective of molecular physics, one could easily envision applications of these ideas to the study of coherence transfer via intramolecular couplings, which are often treated phenomenologically under the secular approximation, i.e. assuming a single decay constant for each coupling process without considering the dynamics of each coherence. Potentially, the same procedure applied here could apply to intramolecular vibrational energy transfer by rovibrational coupling (The Journal of Physical Chemistry 100.31 (1996): 12735-12756), or rotational-electronic coherence transfer (Faraday discussions 194 (2016): 463-478, Physical review letters 111.24 (2013): 243005).

We are grateful to the reviewer for this suggested possible extension of our study that we will investigate in the future.

Given the broad interest and potential applicability of the ideas presented in this manuscript, its publication in Nature Communications is recommended after the following points of interest are addressed.

Point 1: Coherence transfer

Within the secular approximation it is clear that each coherence decays with a time constant related to the corresponding relaxation matrix element. While the behavior of coherence transfer as a function of time has been discussed, it is difficult to envision the time dependent behavior of a single coherence. It would be illuminating to see a plot of one of the coherences addressed in Fig. 4 as a function of time. Perhaps this could be included as a second panel in Fig. 4.

Comment taken into account: Figure 4 has been updated and, as suggested by the reviewer, some coherences, computed with the secular and nonsecular models are now displayed in the layers added below the original figure, **as shown below**

Figure 4: Secular and nonsecular effects. (a) Relative modification of the $\rho_{J_0 M_0, J_0 + 2 M_0}(t)$ coherence for $J_0=15$ induced by collisional transfers from the coherences $\rho_{J_1 M_1, J_1 + 2 M_1}(t)$ with $J_1=17$ (black), $J_1=19$ (red), $J_1=21$ (green), and $J_1=23$ (blue), all normalized to unity at $t=0$. This simple modeling (see Supplementary Information) of coherence transfers considering only few rotational states around the most populated state of N_2O at 300 K reveals that during the short time evolution of the system, exchanges between coherences are efficient, which slow down the decay of the alignment factor with respect to what would be obtained using the secular approximation that neglects these collisional transfers and only takes the losses into account. **(b) and (c)** Computed real and imaginary parts of the $\rho_{JM=0, J+2M=0}(t)$ coherences for N_2O in He at 33 amagat, obtained using the nonsecular (full lines) and secular (dashed lines) models.

The new Figures 4b and 4c are referred to in the text, as explained in the answer to the newcomment (point 2) by the reviewer

Point 2: Further discussion of the loss of coherence due to a high rate of transfer at early times is also required. If I understand eq. (1) correctly, transfer does not imply loss, so how does enhanced transfer lead to loss?

Comment taken into account: Before explaining the changes made to the text, let us mention that, within the secular approximation, Eq. (3) [by the way, we assume that the reviewer above meant Eq. (3) and not Eq. (1)] becomes, for a coherences, i.e. for $i \neq j$, to:

$$\left(\frac{d\rho_{ij}(t)}{dt} \right)_{\text{secular}} = -d\Lambda_{ij,ij}\rho_{ij}(t) \quad (\text{C1})$$

Since all diagonal terms of the relaxation matrix are positive, this equation directly results in a reduction of the amplitude of the coherence and thus a "loss". If one goes beyond the secular approximation, the preceding equation becomes

$$\begin{aligned} \left(\frac{d\rho_{ij}(t)}{dt} \right)_{\text{non secular}} &= -d\Lambda_{ij,ij}\rho_{ij}(t) - d \sum_{(i',j') \neq (i,j)} \Lambda_{ij,i'j'}\rho_{i'j'}(t) \\ &= \left(\frac{d\rho_{ij}(t)}{dt} \right)_{\text{secular}} - d \sum_{(i',j') \neq (i,j)} \Lambda_{ij,i'j'}\rho_{i'j'}(t) \end{aligned} \quad (\text{C2})$$

where the terms with $i' \neq j'$ correspond to the transfer between the $i'j'$ and ij coherences and those with $i'=j'$ to the transfer between the $i'i$ population and the ij coherence. Now it turns out that the second term in the equation above makes, on average, a positive contribution that thus reduces the decay of the ij coherence amplitude. In order to make these points clearer, the text was modified as explained below:

- The original text on page 3 "... without quenching it. As shown by the simulations represented by the black line in Fig. 3, ..." was replaced by: "... without quenching it. Under these approximations, the equation driving the decay of the coherences (i.e. for $i \neq j$) becomes

$$\left(\frac{d\rho_{ij}(t)}{dt} \right)_{\text{Coll}}^{\text{Secular}} = -d\Lambda_{ij,ij}\rho_{ij}(t) \quad . \quad (4)$$

The diagonal elements $\Lambda_{ij,ij}$ have been estimated for N_2O -He at room temperature (see the Supplementary Information) using the Infinite Order Sudden Approximation (IOSA). The calculations show that all these terms are positive, practically independent of i and j and equal to $0.013 \text{ ps}^{-1} \cdot \text{amagat}^{-1}$. This means that the echo decay within the secular approximation is independent of the delay. As shown by the simulations represented by the black line in Fig. 3, the secular theory well reproduces the observed decays of the echoes in the plateau region beyond about $2\tau_{12} = 10 \text{ ps}$ with a density normalized time constant of $1/0.013 = 76 \text{ ps} \cdot \text{amagat}$. In contrast, the secular model does not capture the weaker efficiency of collisions at shorter times and the fact that the decrease of the echo is here all the more slow when the delay is short". In addition, we hope that the additional changes made and described in the answer to the next point just below make things clearer

Point 3: Density-Normalized Characteristic Time Constant

The time constant as a function of delay extracted from the experimental and simulated data show that coherence transfer is important at early times. However, no direct connection is

made between this time constant and the relaxation matrix, or any other fundamental theoretical parameter. Can the authors comment on the connection between this measured quantity and a parameter of the quantum system?

Comment taken into account: In the case of the secular model, explained just above, the connection between the decay of the coherences and the alignment echo amplitude is direct and we hope that the answer to the preceding comment makes things clear in this case. When a nonsecular model is used, the time evolution of the density matrix is given by:

$$\left(\frac{d\rho_{ij}(t)}{dt}\right)_{\text{Coll}}^{\text{Non secular}} = \left(\frac{d\rho_{ij}(t)}{dt}\right)_{\text{Coll}}^{\text{Secular}} - d \sum_{(i',j') \neq (i,j)} \Lambda_{ij,i'j'} \rho_{i'j'}(t)$$

Analysis shows that the effect of the various contributions in the sum on the right hand side is to reduce the collision-induced decay of the echo amplitude. In order to mention these elements, the following changes have been made.

- The text *"In order to go further, computations of the echo decays have been carried by solving equations (1)-(3) now keeping all collisional transfers channels and thus accounting for secular and nonsecular terms. The required relaxation matrix elements have been constructed ..."* was changed to *"In order to go further, computations of the echo decays have been carried by solving equations (1)-(3) now keeping all collisional transfers channels and thus accounting for secular and nonsecular terms. Equation (3) thus becomes:*

$$\left(\frac{d\rho_{ij}(t)}{dt}\right)_{\text{Coll}}^{\text{Non secular}} = \left(\frac{d\rho_{ij}(t)}{dt}\right)_{\text{Coll}}^{\text{Secular}} - d \sum_{(i',j') \neq (i,j)} \Lambda_{ij,i'j'} \rho_{i'j'}(t), \quad (6)$$

where the first term on the right-hand side is given by equation (4). The required relaxation matrix elements have been constructed ..."

- Further down, the text was changed to *"Figure 3 confirms that the nonsecular terms [within the sum on the right-hand side of equation (6)], which describe the transfers among coherences as well as between populations and coherences, globally contribute to reduce the pressure-induced echo decay at short times. Analysis shows that this results from the fact that these contributions here systematically reduce the collision-induced decay of the amplitudes of the $\rho_{JM,J\pm 2M}(t)$ coherences, as exemplified in Figs. 4b and c, and thus also of the echo. In particular, for N₂O diluted in He, the IOS model predicts that, immediately after the first pulse, nonsecular couplings between coherences reduce the collisional damping of these amplitudes by a factor of about two with respect to the secular behavior. Neglecting the nonsecular terms thus leads to overestimate the decay of the system. Note that a similar pattern was observed in Ref. 24 in the modeling of decoherence and dissipation in non adiabatic cis-trans photoisomerization. As time goes by, the averaging out of these terms, due to progressive dephasing of the coherences, (see Fig. 4a) results..."*

Reviewer #2 (Remarks to the Author):

The manuscript by J.Ma et.al provides a clear and elegant experimental observation for the non-markovian collisional relaxation of molecular rotational motion. The authors show that on short time scales (<10ps) the secular / markovian treatment for the collision-induced decay is incorrect and that the "collision efficiency" is lower than at longer time scales in which the "memory" is lost and the secular/markovian approximation is valid. These findings show that at short time scales the effect of collisions is very different from the common treatment of collisional relaxation done in the past. Rotational dynamics on such short time scale (that are much shorter than the rotational period) were previously unattainable in rotational dynamics experiments (with which the authors have long experience and numerous important contributions to) and have become feasible only recently with the development of rotational echo measurements (a technique that some of the authors are among its pioneers [e.g. PRL. 114. 153601 (2015)]).

The topic is very timely, the results are uniquely original and utilize novel methodologies. The experimental results are clean and very persuasive. The results are clearly presented and are fully supported by state-of-the-art numerical and theoretical calculations including an elaborated SI section.

It is with great pleasure that I support acceptance and publication of this work in Nature Communications.

Few comments:

Point 1: equation (2): the authors probably meant to write $\cos^2(\theta)$ without the $-1/3$.

Comment taken into account: We thank the referee for his remark. In fact, equation (2) is missing a term proportional to $\bar{\alpha} = (2\alpha_{\perp} + \alpha_{\parallel})/3$, where $\bar{\alpha}$ is the average polarizability of the molecule. The correct expression should write

$$\mathbf{H}_L(t) = -\frac{1}{4}E^2(t) \left[\bar{\alpha} + \Delta\alpha \left(\cos^2\theta - \frac{1}{3} \right) \right]$$

The term $\bar{\alpha} - \Delta\alpha/3 = \alpha_{\perp}$, responsible for a Stark shift of all the rotational states by the same amount, is generally omitted for the reason that it does not participate to the alignment dynamics. For this reason, we have **replaced equation 2 by**

$$\mathbf{H}_L(t) = -\frac{1}{4}E^2(t)\Delta\alpha\cos^2\theta$$

and specified in the text that this equation defines the Hamiltonian **"(up to an irrelevant term that does not multiply the direction cosine operator $\cos^2\theta$)"**

Point 2: It would aid the readers if the authors note that they refer to the delay of 4.8 ps when they discuss the short time scale of up to 10 ps.

Comment taken into account: "... beyond about 10 ps ..." was changed to "... beyond about $2\tau_{12} = 10$ ps ..."

Point 3: The authors write: "As shown by the simulations represented by the black line in Fig. 3, the secular theory well reproduces the observed decays of the echoes in the plateau region beyond 10 ps, but does not capture the weaker efficiency of collisions and the variation of the echo decay at short times." - I would suggest that the authors expand this very central statement by walking the readers through the results.

Comment take into account: These sentences were changed to "*As shown by the simulations represented by the black dashed line in Fig. 3, the secular theory well reproduces the observed decays of the echoes in the plateau region beyond about $2\tau_{12}=10$ ps with a density normalized time constant of $1/0.013=76$ ps.amagat. In contrast, the secular model does not capture the weaker efficiency of collisions at shorter times and the fact that the decrease of the echo is here all the more slow when the delay is short.*"

Also please see our response to the last question raised by the first Reviewer as well as the corresponding modifications made in manuscript.

Point 4: last sentence of the first SI paragraph: 'amond' should be 'among'

Comment taken into account

Suggestions/questions (for the authors to decide whether to address or not)

Point 5: If possible, it would be nice to include an 'easy to understand' classical explanation about the nature of the non-Markovian collisions, i.e. how does the collision of the N₂O with the Helium atoms creates "memory effects in the system-bath "history"".

Comment taken into account: We are not sure that we properly understand this comment, because, as was stated in the original text, just above Eq. (3), "*Assuming a Markovian model of collisions (valid here because N₂O and He interact very shortly and in a very limited range of intermolecular distance²⁰)*". Hence, giving an explanation on non-Markovian effects is not, strictly speaking, relevant for the present study. We however agree that it is of interest in order to broaden the discussion and we made the following two changes to the text:

-The sentence "*Assuming a Markovian model of collisions (valid here because N₂O and He interact very shortly and in a very limited range of intermolecular distance²⁰) and ...*," appearing just above Eq. (3) was change to "*Assuming a Markovian model of collisions (valid here because, as discussed in the Supplementary Information, N₂O and He interact very shortly and in a very limited range of intermolecular distance²⁰), and ...*"

- The following section was added at the end of the Supplementary Information file:

"On the Markovian approximation

The Markovian approximation for rotational relaxation and decoherence processes assumes that the quantity of interest (here the alignment factor, through the time evolution of the density matrix) can be modeled by considering that all collisions are complete. Also called "impact" approximation it thus neglects the "memory effects" associated with those collisions that are on-going at the time ($t=0$) of the excitation of the system or at the time ($t=2\tau_{12}$, for the echo) where the system is observed. This implicitly assumes that the relative number of on-going collisions is very small and/or that they have an effect over a very short interval of time. For the system of N₂O gas highly diluted in He investigated in the present study, both criteria are well satisfied so that the Markovian approximation is valid. Indeed, the N₂O-He accurate ab initio intermolecular potential [16] has an almost negligible well (with respect to the mean kinetic energy at room temperature). The repulsive front is relevant (i.e. has values between 0 and 600 K) in the $[R_0-\Delta R, R_0]$ range of intermolecular distances where R_0 varies between 2.7 and 4.2 Å depending on the respective orientations of the molecular and intermolecular axes and ΔR is typically 0.5 Å. This means that, for a N₂O molecule to be colliding with a He atom, the latter should be in the interval between the two spheres centered on the N₂O center of mass and of typical radii 3.5 to 3.0 Å. This corresponds to a very small volume (about 70 Å³) that statistically contains very few He

atoms unless the gas density is very high [for 15 amagat which corresponds to 4×10^{26} atom/m³, this statistical number is about 0.025]. In addition, the mean relative speed of N₂O-He collisions at 295 K is 1300 m/s which means that the distance between the two spheres (0.5 Å) is, on average, traveled in 0.038 ps. The above given numbers show that non-Markovian effects are of very small amplitude and only exist at time scales much shorter than all those involved in the present study. Assuming Markovian collisions is thus a very good approximation for the N₂O-He system but this would not be the case for pairs of colliders involving much smaller relative speeds and intermolecular forces at much longer ranges."

Point 6: Does the coherence amplitude (namely if the experiment is conducted with different intensities) play a role? will the memory effect hold for longer/shorter time scales? or maybe not affected at all?

The intensity effects raised by the reviewer is somehow a "technical" as the answer given below indicates that the results are practically insensitive to the intensities of the pulses for the intensities used in the present work. The following text answer the reviewer's questions: Test calculations show that, for laser pulse peak intensities comparable to those used in the present experiments, the amplitude of the echo is proportional to the product of the intensity of the first pulse and of the squared intensity of the second pulse, a behavior confirmed by experiments (see [Rosenberg et al, Phys. Rev. Lett. **121**, 234101 (2018)] and [Rosenberg et al, J. Phys. Chem. Lett. **8**, 5128 (2017)]). For the coherences, it is relatively easy to show that the amplitude of the $\langle J, M | \rho | J + \Delta J, M \rangle$ matrix element generated by a pulse is proportional to $I^{|\Delta J|/2}$ with I the intensity of the pulse. Despite these dependences, the decay of a given echo (i.e. a given value of τ_{12}) with increasing gas density is, unless larger intensities than 50 TW/cm² are used, almost insensitive to these intensities. The reason for this is that the energy deposited in the system through the change of the populations of the rotational levels is relatively small and thus negligibly affects, after rotation-translation transfers which increase the translational speed of the molecules, the frequency of collisions. In other words, at moderate intensities, the rotational populations and the distribution of translational velocities remain close to the (initial) Boltzmannian ones. On the opposite, for large intensities, the energy deposited in the system can become large when compared to the initial temperature with a significant enhancement of the populations of high J levels. Part of this energy will then be transferred to translation, with a progressive increase of the kinetic temperature and thus of the frequency of collisions. In this case, the decay of the echo amplitude will be faster. Such an effect was demonstrated, by looking at the decay of revivals, in [Houzet et al, Phys. Rev. A **86**, 033419 (2012)] where it was experimentally and theoretically shown that "preheating" CO₂ gas by a 60 TW/cm² pulse reduces the time constant of the revivals decay by about 7%.

Reviewer #3 (Remarks to the Author):

The paper by Faucher et al is certainly an interesting and very useful method to investigate rotational coherences on very short time scales, especially at time scales of collisions. The emphasis on studying nonsecular effects as compared to previous standard relaxation methods is informative. The paper is therefore publishable following clarification and additional information.

Point 1: the laser intensities used are of the order of 10^{13} Watts/cm². Such intensities

do produce important Stark shifts. The magnitude of these shifts is never stipulated and not clearly identified as relevant or not.

There are two terms in the interaction Hamiltonian that account for Stark effects. The first one,

$-\frac{1}{4}E^2(t)\left(\bar{\alpha}-\frac{\Delta\alpha}{3}\right)=-\frac{1}{4}E^2(t)\alpha_{\perp}$, which was already mentioned in our response to the second

Reviewer, shifts the rotational states by the same amount of energy since the 800 nm radiation of the pump is far detuned from any resonances. So this term can be neglected in the calculations, as it has no influence over the rotational dynamics. The second term,

$-\frac{1}{4}E^2(t)\Delta\alpha\cos^2\theta$, which is responsible for the alignment of the molecular axis, produces a

state-dependent light shift of the rotational levels during their impulsive excitation by the pump laser. Since this effect is inherently taken into consideration in the numerical simulations of the alignment signal and since this is a rather "technical" point, we have not modified the content of the paper to discuss this issue

Point 2: at the pressures used, the time for collisions is not clearly identified

Comment taken into account: The mean time interval between collision is now given in the text. It is firstly mentioned in at the end of the last paragraph before Equation (1), where we added the text: "**Note, for comparison, that the mean time interval between successive N₂O-He collisions, obtained for the intermolecular distance of 3.5 Å where the N₂O-He potential is significant (see the end of the Supplementary Information), is of 70 ps.amagat.**" In addition, in the conclusion, the sentence "*The latter enable to probe the system very soon after its stimulation by two laser kicks, on a time scale that is comparable to the time interval between successive collisions at the high densities used in the measurements*" was changed to "*The latter enable to probe the system very soon after its stimulation by two laser kicks, on a time scale of a few ps that is comparable to the time interval between successive collisions (about 3.5 ps for the density of 20 amagat typical of the experiments) at the high densities used in the measurements*"

Point 3: The laser-molecule interaction assumes no dipole moment. Thus can only be applied to symmetric molecules.??

Comment taken into account: We first recall, because the reviewer's remark may imply that she/he thinks N₂O as a symmetric molecule, that the geometry of N₂O being N-N-O this molecule does have a permanent dipole moment. Now, to answer the reviewer's question, the answer is no, a laser-molecule interaction Hamiltonian only taking into account the molecule polarizability tensor is valid for any type of molecule as long as the period of oscillation of the laser electric field is small enough to be able to consider that the molecule rotation is frozen at this time scale. This is because $\bar{\mu}(t)\cdot\vec{E}(t)=\cos\theta(t)E(t)$ averages to zero when integrated over a period of the field since $\cos\theta(t)$ is constant over such a time interval. For the non-resonant excitation at 800 nm used in the experiment, this period is 2.7 fs which is orders of magnitude smaller than the rotational speed associated with the rotational levels all molecules that are significantly populated at room temperature. Hence, the oscillating term $\bar{\mu}(t)\cdot\vec{E}(t)$ cancels out when integrated over one period of oscillation of the field during which the molecule does not rotate and $\bar{\mu}(t)$ can be considered as time independent. Now, of course, if the frequency of the field is close to a resonance of the system, which is the case for a THz excitation, the situation is different and the molecules can then not only be aligned but also oriented through their permanent electric dipole. In order to include such elements in the paper in a concise way (since these remarks are somehow "on the side" from the main focus of the present study), we made the following changes.

- Below Eq. (2) we added the following sentences "*Note that, in principle, the contribution of the permanent dipole moment of N₂O should be included in equation (2). However, since the excitation field used in the experiment (at 800 nm) oscillates with a period (2.7 fs) much smaller than that of the rotation of the molecule for the thermally populated levels, the associated contribution to equation (2) averages out.*".

- In addition, since the present study that uses the alignment echoes to investigate decoherence at early times is not limited neither to linear nor symmetric molecules and not limited to non-resonant excitations, the following sentences were added at the end of the conclusion paragraph. "**Finally, note that equivalent studies could be made for other types of molecules (including polar and nonlinear ones) and/or carried out by using a different process in order to excite the system at $t=0$ and $t=\tau_{12}$ and generate an echo at $2\tau_{12}$. For instance, early time decoherence could be investigated for CH₃I using THz excitations as done in Refs. [24-26] with a single pulse, which is only one example among many others of the great potentialities of the approach introduced in this paper to study early time dissipation.**"

Point 4: as shown previously by Matta et al in J Chem Phys 121,7764 (2006) a permanent dipole moment and polarizability correspond to a total HL comprised of a static term and a $w,2w$ time dependent interaction. Since the pump is at 800nm and the probe at 400nm, this would suggest interference effects from both pulses in the polarizability. Has this been observed?

Comment disregarded: As explained in the answer to Point 3, the coupling of the field with the permanent dipole moment of the molecule is negligible in the present system because the 800 nm radiation of the pump is far detuned from the resonances of the system [which is not the case in the reference mentioned by the reviewer, i.e. J Chem Phys 121,7764 (2004)]. Additionally, the intensity of the probe pulse at 400 nm is kept low in order to avoid any nonlinear coupling with the system through the polarizability. As a result, the weak 400 nm probe does not influence the rotational dynamics of the system.

Point 5: The imaginary part of equn 4 would contain relevant Stark and collisional energy shifts. Has this been included in the simulations ?

Comment taken into account: As explained in the answer to Point 1, the J-dependent Starks shifts of the rotational states are implicitly included in the interaction Hamiltonian defined by Eq. (2). Regarding Eq. (5), corresponding to the previous equation (4), the effect depicted in Figure 4(a) for the real part of $\Delta\rho/\rho$ also stands for the imaginary part of $\Delta\rho/\rho$. Additionally, no collisional shift is considered in the present work since all relaxation relaxation matrix elements are purely real.

Below Eq. (5) we added the following sentences "**Note that the imaginary part of $\Delta\rho/\rho$ exhibits the same behavior.**"

REVIEWERS' COMMENTS:

Reviewer #1 (Remarks to the Author):

I would like to thank the authors for careful consideration of my comments. In the previous version of the manuscript it was not clear to me that efficient coherence transfer in fact reduces the efficacy of collisional decoherence. The inclusion of the discussion around eq.6 and Fig. 4 make this abundantly clear in the new version of the manuscript.

I recommend publication of the manuscript in its current form.

Reviewer #2 (Remarks to the Author):

The authors have addressed all of the concerns and remarks (made by all of the reviewers) and have made the necessary changes in the text.
I support publication of this work in Nature communication in its current form.

Reviewer #3 (Remarks to the Author):

Excellent work and revision.

NCOMMS-19-26218A

Title: Observing collisions beyond the secular approximation limit"

NO NEW ISSUES RAISED BY THE REFEREES